# Transfer of Deep Reactive Policies for MDP Planning

**Aniket Bajpai, Sankalp Garg, Mausam**
Indian Institute of Technology, Delhi
New Delhi, India
{quantum.computing96, sankalp2621998}@gmail.com, mausam@cse.iitd.ac.in

## Abstract

Domain-independent probabilistic planners input an MDP description in a factored representation language such as PPDDL or RDDL, and exploit the specifics of the representation for faster planning. Traditional algorithms operate on each problem instance independently, and good methods for transferring experience from policies of other instances of a domain to a new instance do not exist. Recently, researchers have begun exploring the use of deep reactive policies, trained via deep reinforcement learning (RL), for MDP planning domains. One advantage of deep reactive policies is that they are more amenable to transfer learning.

In this paper, we present the first domain-independent transfer algorithm for MDP planning domains expressed in an RDDL representation. Our architecture exploits the symbolic state configuration and transition function of the domain (available via RDDL) to learn a shared embedding space for states and state-action pairs for all problem instances of a domain. We then learn an RL agent in the embedding space, making a near zero-shot transfer possible, i.e., without much training on the new instance, and without using the domain simulator at all. Experiments on three different benchmark domains underscore the value of our transfer algorithm. Compared against planning from scratch, and a state-of-the-art RL transfer algorithm, our transfer solution has significantly superior learning curves.

## 1 Introduction

The field of domain-independent planning designs planners that can be run for all symbolic planning problems described in a given input representation. The planners use representation-specific algorithms, thus allowing themselves to be run on all domains that can be expressed in the representation.

Two popular representation languages for expressing probabilistic planning problems are PPDDL, Probabilistic Planning Domain Description Language [Younes et al., 2005], and RDDL, Relational Dynamic Influence Diagram Language [Sanner, 2010]. These languages express Markov Decision Processes (MDPs) with potentially large state spaces using a factored representation. Most traditional algorithms for PPDDL and RDDL planning solve each problem instance independently are not able to share policies between two or more problems in the same domain [Mausam and Kolobov, 2012].

Very recently, probabilistic planning researchers have begun exploring ideas from deep reinforcement learning, which approximates state-value or state-action mappings via deep neural networks. Deep RL algorithms only expect a domain simulator and do not expect any domain model. Since the RDDL or PPDDL models can always be converted to a simulator, every model-free RL algorithm is applicable to the MDP planning setting. Recent results have shown competitive performance of deep reactive policies generated by RL on several planning benchmarks [Fern et al., 2018, Toyer et al., 2018].

Because neural models can learn latent representations, they can be effective at efficient transfer from one problem to another. In this paper, we present the first domain-independent transfer algorithm for

MDP planning domains expressed in RDDL. As a first step towards this general research area, we investigate transfer between *equi-sized* problems from the same domain, i.e. those with same state variables, but different connectivity graphs.

We name our novel neural architecture, TORPIDO – **T**ransfer **o**f **R**eactive **P**olicies **I**ndependent of **Do**mains. TORPIDO brings together several innovations that are well-suited to symbolic planning problems. First, it exploits the given symbolic (factored) state and its connectivity structure to generate a state embedding using a graph convolutional network [Goyal and Ferrara, 2017]. An RL agent learnt over the state embeddings transfers well across problems. One of the most important challenges for our model are the actions, which are also expressed in a symbolic language. The same ground action name in two problem instances may actually mean different actions, since the state variables over which the action is applied may have different interpretations. As a second innovation, we train an action decoder that learns the mapping from instance-independent state-action embedding to an instance-specific ground action. Third, we make use of the given transition function by training an instance-independent model of domain transition in the embedding space. This gets transferred well and enables a fast learning of action decoder for every new instance. Finally, as a fourth innovation, we also implement an adversarially optimized instance classifier, whose job is to predict which instance a given embedding is coming from. It helps TORPIDO to learn instance-independent embeddings more effectively.

We perform experiments across three standard RDDL domains from IPPC, the International Probabilistic Planning Competition [Grzes et al., 2014]. We compare the learning curves of TORPIDO with A3C, a state of the art deep RL engine [Mnih et al., 2016], and Attend-Adapt-Transfer (A2T), a state of the art deep RL transfer algorithm [Rajendran et al., 2017]. We find that TORPIDO has a much superior learning performance, i.e, obtains a much higher reward for the same number of learning steps. Its strength is in near-zero shot learning – it can quickly train an action decoder for a new instance and obtains a much higher reward than baselines without running any RL or making any simulator calls. To summarize,

1. We present the first domain-independent transfer algorithm for symbolic MDP domains expressed in RDDL language.

2. Our novel architecture TORPIDO uses the graph structure and transition function present in RDDL to induce instance-independent RL, state encoder and other components. These can be easily transferred to a test instance and only an action decoder is learned from scratch.

3. TORPIDO has significantly superior learning curves compared to existing transfer algorithms and training from scratch. Its particular strength is its near-zero shot transfer – transfer of policies without running any RL.

We release the code of TORPIDO for future research.[1]

## 2 Background and Related Work

### 2.1 Reinforcement Learning

In its standard setting, an RL agent acts for long periods of time in an uncertain environment and wishes to maximize its long-term return. Its dynamics is modeled via a Markov Decision Process (MDP), which takes as input a state space $S$, an action space $A$, unknown transition dynamics $Pr$, and an unknown reward function $R$ [Puterman, 1994]. The agent in state $s_t$ at time $t$ takes an action $a_t$ to get a reward $r_t$ and make a transition to $s_{t+1}$ via its MDP dynamics. The $h$-step return $R_{t:t+h}$ is defined as the discounted sum of rewards, $\sum_{i=1}^{h} \gamma^{i-1} r_{t+i}$. The value function $V_\pi(s)$ is the expected (infinite step) discounted return from state $s$ if all actions are selected according to policy $\pi(a|s)$. The action value function $Q_\pi(s, a)$ is the expected discounted return after taking action $a$ in state $s$ and then selecting actions according to $\pi(a|s)$ thereafter.

Deep RL algorithms approximate policy [Williams, 1992] or value function [Mnih et al., 2015] or both [Mnih et al., 2016] via a neural network. Our work builds upon the Asynchronous Advantage Actor-Critic (A3C) algorithm [Mnih et al., 2016], which constructs approximations for both the

policy (using the 'actor' network) and the value function (using the 'critic' network). The parameters of the critic network are adjusted to maximize the expected reward by using the gradient of the 'advantage' function, which measures the improvement of the action over the expected state value $A(s, a) = Q(s, a) - V(s)$. Hence the update to the critic network is the expectation of $\frac{\partial}{\partial\theta} log\pi(a|s)(Q_\pi(s, a; \theta) - V(s; \theta))$. The actor network maximizes the $H$-step lookahead reward by minimizing the expectation of mean squared loss, $(R_{t:t+H} + \gamma^H V(s_{t+H+1}; \theta^-) - V(s_t; \theta))^2$. Here, the optimization is with respect to $\theta$, the cumulative parameters in both actor and critic networks, and $\theta^-$ are their previous values. Furthermore, many instances of the agent interact in parallel with many instances of the environment, which both accelerates and stabilizes learning in A3C.

**Transfer Learning in Deep RL:** Neural models are highly amenable for transfer learning, because they can learn generalized representations. Initial approaches to transfer learning in deep RL involved transferring the value function or policies from the source to the target task. More recent methods have developed these ideas further, for example, by using expert policies from multiple tasks and combining them with source task features to learn an actor-mimic policy [Parisotto et al., 2015], or by using a teacher network to propose a curriculum over tasks for effective multi-task learning [Matiisen et al., 2017]. A preliminary approach has also used a symbolic front-end for deep RL [Garnelo et al., 2016]. We compare our transfer algorithm against a recent algorithm that uses an attention mechanism to allow selective transfer and avoid negative transfer [Rajendran et al., 2017].

Recently, there have also been attempts at performing a zero-shot transfer, i.e., without seeing the new domain. An example is DARLA [Higgins et al., 2017], which leverages the recent work on domain adaptation to learn a domain-independent representation of the state, and learns a policy over this state representation, hence making the learned policy robust to domain shifts. Our work attempts a near-zero shot learning by learning a good policy with limited learning, and without any RL.

## 2.2 Probabilistic Planning

Planning problems are special cases of RL problems in which the transition function and reward function are known [Mausam and Kolobov, 2012]. In this work, we consider planning problems that model finite-horizon discounted reward MDPs with a known initial state [Kolobov et al., 2012]. Thus, our problems take as input $\langle S, A, Pr, R, H, s_0, \gamma \rangle$, where $H$ is the horizon for the problem. Probabilistic planners can use the model to perform a full Bellman backup, i.e., expectation over all next outcomes from an action (e.g., in Value Iteration [Bellman, 1957]), whereas RL agents can only backup from a single sampled next state (e.g., in Q Learning [Sutton and Barto, 1998]).

Factored MDPs provide a more compact way to represent MDP problems. They decompose a state $s$ into a set of $n$ binary state variables $(x_1, x_2, \ldots, x_n)$; the transition function specifies the change in each state variable, and the reward function also uses a factored representation. Solving a finite-horizon factored MDP is EXPTIME-complete, because it can represent an MDP exponential states in polynomial size [Littman, 1997].

**RDDL Reprentation:** RDDL describes a factored MDP via objects, predicates and functions. It is a first-order representation, i.e., it can be initiated with a different set of objects to construct MDPs from the same domain. A domain has parameterized *non-fluents* to represent the part of the state space that does not change. A planning state needs those state variables that can change via actions or natural dynamics. They are represented as parameterized *fluents*. The transition function for the system is specified via (stochastic) functions over next state variables conditioned on current state variables and actions. The reward function is also defined in the factored form using the state variables.

We illustrate the RDDL language via the SysAdmin domain [Guestrin et al., 2001], which consists of a set of $K$ computers connected in a network. Each computer in the network can be shut down with a probability dependent on the ratio of its 'on' neighbours to total number of neighbours. Any 'off' computer can randomly switch on with a reboot probability. The agent can take the action of rebooting a single computer, or no action at all in each time step. Note that no-op is also a valid action as this domain would evolve even if the agent does not take any action. The reward at each timestep is the number of 'on' computers at that timestep. The natural structure of the problem, and the factored nature of the transition and reward function make this problem perfectly suited to be represented in RDDL as follows. Objects: $c_1, ...c_K$, the $K$ computers; Non fluents: $connected(c_j, c_i)$, whose value is 1 if $c_j$ is a neighbor of $c_i$; State fluents: $on(c_i)$, which is 1 if the $i^{th}$ computer is on; Action fluents: $reboot(c_i)$, which denotes that the agent rebooted the $i^{th}$ computer;

Reward function: $\Sigma i[on(c_i)]$; Transition function: If $reboot(c_i)$, then $on'(c_i) = 1$, Else if $on(c_i)$ then $on'(c_i) = Bernoulli(a+b \times (1+\Sigma j \mathbb{1}[connected(c_j,c_i) \wedge on(c_j)])/(1+\Sigma j \mathbb{1}[connected(c_j,c_i)]))$, Else $on'(c_i) = Bernoulli(d)$. Here all primed fluents denote the value at the next time step, and $a$, $b$, and $d$ are constants modeling the dynamics of the domain.

**Deep Learning for Planning:** Value Iteration Networks formalize the idea of running the Value Iteration algorithm within the neural model [Tamar et al., 2017], however, they operate on the state space, instead of factored space. There have been three recent works on the use of neural architectures for domain-independent factored MDP planning. One work learns deep reactive policies by using a network that mimics the local dependency structure in the RDDL representation of the problem [Fern et al., 2018]. We are also interested in problems specified in RDDL, but are more focused on transfer across problem instances. There also has been an early attempt on transfer in planning problems, for two specific classical (deterministic) domains of TSP and Sokoban [Groshev et al., 2018]. In contrast, we propose a transfer mechanism that can be used for domain-independent RDDL planning. Finally, Action-Schema Nets use layers of propositions and actions for solving and transferring between goal-oriented PPDDL planning problems [Toyer et al., 2018]. RDDL allows concurrent conditional effects, which when converted to PPDDL can lead to exponential blowup in the action space. Therefore, ASNets are not scalable to RDDL domains considered in this paper.

### 2.3 Graph Convolutional Networks

Graph Convolutional Networks (GCN) generalize convolutional networks to arbitrarily structured graphs [Goyal and Ferrara, 2017]. A GCN layer take as input a feature representation for every node in the graph ($M \times D_I$ feature matrix, where $M$ is the number of nodes in the graph, and $D_I$ is the input feature dimension), and a representation for the graph structure (an $M \times M$ adjacency matrix $A$), and produces an output feature representation for every node (in the form of an $M \times D_O$ matrix, where $D_O$ is the output feature dimension). A layer of GCN can be written as $F^{(l+1)} = g(F^{(l)}, A)$. where $F^{(l)}$ and $F^{(l+1)}$ are the feature representations for $l^{th}$ and $(l+1)^{th}$ layers.

We use this propagation rule (from [Kipf and Welling, 2017]) in our work: $g(F^{(l)}, A)) = \sigma(\hat{D}^{-12}\hat{A}\hat{D}^{-12}F^{(l)}W^{(l)})$, where $\hat{A} = A + I$, $I$ being the identity matrix, and $\hat{D}$ is the diagonal node degree matrix of $\hat{A}$. Intuitively, this propagation rule implies that the feature at a particular node of the $(l+1)^{th}$ layer is the weighted sum of the features of the node and all its neighbours at the $l^{th}$ layer. Furthermore, these weights are shared at all nodes of the graph, similar to how the weights of a CNN kernel are shared at all locations of the image. Hence, at each layer, the GCN expands its receptive field at each node by 1. A deep GCN network, i.e., a network consisting of stacked GCN layers, can therefore have a large enough receptive field and construct good feature representations for each node of a graph.

## 3 Problem formulation

The transfer learning problem is formulated as follows. We wish to learn the policy of the target problem instance $P_T$, where in addition to $P_T$, we are given $N$ source problem instances $P_1, P_2, ..., P_N$. For this paper, we make the assumption that the state, action spaces and rewards of all problems is the same, even though their initial state, and non-fluents could be different. For example, computers in SysAdmin may be arranged in different topologies (based on different values of non-fluents *connected*). Any transfer learning solution will operate in two phases: (1) **Learning phase:** Learn policies $\pi_1, \pi_2, ..., \pi_N$ over each source problem instance, and possibly also learn general representations that will help in transfer. (2) **Transfer phase:** Learn the policy $\pi_T$ for $P_T$ using all output of the learning phase.

An ideal zero-shot transfer approach will be one where the target instance environment will not even be used during the transfer phase. Two indicators of good transfer are a high pre-train (zero-shot transfer) score, and a more sample-efficient learning compared to a policy learnt from scratch on $P_T$.

## 4 Transfer Learning framework

Our approach hinges on the claim that there exists a 'good' embedding space for all states, as well as a 'good' embedding space of all state-action pairs, which is shared by all equi-sized instances of a

given domain. A 'good' state embedding space is a space in which similar states are close together and dissimilar states are far apart (similarly for state-action pair embeddings).

Our neural architecture is shown in Figure 1. Broadly, our architecture has five components: a state encoder (SE), an action decoder (SAD), an RL module (RL), a transition module (Tr), and an instance classifier (IC). In the training phase, TORPIDO learns instance-independent modules for SE, RL, Tr and IC, but an instance-specific $SAD^I$ (for $P_I$). Its transfer phase operates in two steps. First, using the general SE and Tr models, it learns weights for $SAD^T$, the target action decoder. We call this near-zero shot learning, because this can compute a policy $\pi_T$ for $P_T$ without running any RL. Once, this $SAD^T$ is effectively trained, we transfer all other components and retrain them via RL to improve the policy further for the target instance.

**State Encoder:** We leverage the structure of RDDL domains to represent the instance in the form of a graph (state variables as nodes, edges between nodes if the respective objects are connected via the non-fluents in the domain). The state encoder takes the adjacency matrix for the instance graph and the current state as input, and transforms the state to its embedding. We use the Graph Convolution Network (GCN) to perform this encoding. The GCN constructs multiple features for each node at each layer. Hence, the output of the deep GCN consists of multi-dimensional features at each node, which represent embeddings for the corresponding state variables. These embeddings are concatenated to produce the final state embedding, which is the output of the state encoder module.

**RL Module:** This deep RL agent takes in a state embedding as input and outputs a policy in the form of a state-action embedding. This embedding is an abstract representation of a *soft action*, i.e., a distribution over actions in the embedding space (which will be further decoded by action encoder). We think of this embedding as representing the *pair* of state and soft action, instead of just a soft action. This is because the same action may have different effects depending on the state, and hence a standalone action embedding would not make sense. This can be seen as a neural representation for the notion of state-action pair symmetries in RL domains [Anand et al., 2015, 2016]. We use the A3C algorithm to learn our RL agent, because it has been shown to be robust and stable [Mnih et al., 2016], though other RL variants can easily replace this module. A3C uses a simulator, which can be easily created by sampling from the known transition function in the RDDL representation.

We note that only the policy network of the A3C agent is shared between instances and operates in the embedding space. The value network is different for each instance and operates in the original state space. We did not try to learn a transferable value function as we were ultimately only concerned with the policy, and not the value in the target domain. Hence, in our case, the sole purpose of the value function is to assist the policy function in learning a good policy.

**Action decoder:** The action decoder aims to learn a transformation from the state-action embedding to a soft action (probability distribution over actions). However, such a transformation would not be well-defined, as a state-action embedding could correspond to more than one (symmetric) state-action pairs, and hence more than one corresponding actions. E.g., consider a navigation problem over a square grid [Ravindran, 2004], with the goal at top-right corner. Its immediate neighbors (the state to the left say $s_1$, and state below, say $s_2$) will be symmetric as they both can reach the goal in one step. We expect them to have the same state-action pair embedding with their respective optimal actions. However, the optimal actions are "right" for $s_1$ and "up" for $s_2$.

To resolve this ambiguity, we need to input the state as well into the action decoder. The decoder outputs a probability distribution over actions $\pi(s)$. TORPIDO implements the action decoder using a fully connected network with a softmax to output a distribution over actions. It is important to realize that we need a separate action decoder for each instance, as the required transformation is different for different instances. For example, if the navigation problem is changed so that the goal is now in the lower left corner, all other embeddings may transfer, but the final action output will be different ("down" and "left" for states symmetric to $s_1$ and $s_2$). Action decoder is the only component, which cannot be directly transferred from source problems to the target problem.

**Transition Transfer Module:** To speed up the transfer to the target domain, we additionally learn a transition module in the learning phase. This module takes in as input the current and next state embeddings $(s, s')$, and outputs a soft-action embedding (interpreted as a distribution over actions), with the semantics that the output distribution $p(a) = \frac{Pr(s,a,s')}{\sum_{a'} Pr(s,a',s')}$. I.e., the output embedding maintains which action is more likely to be responsible for the transition from $s$ to $s'$. The gold data for training this module can be easily computed via the RDDL representation. Note that the transition

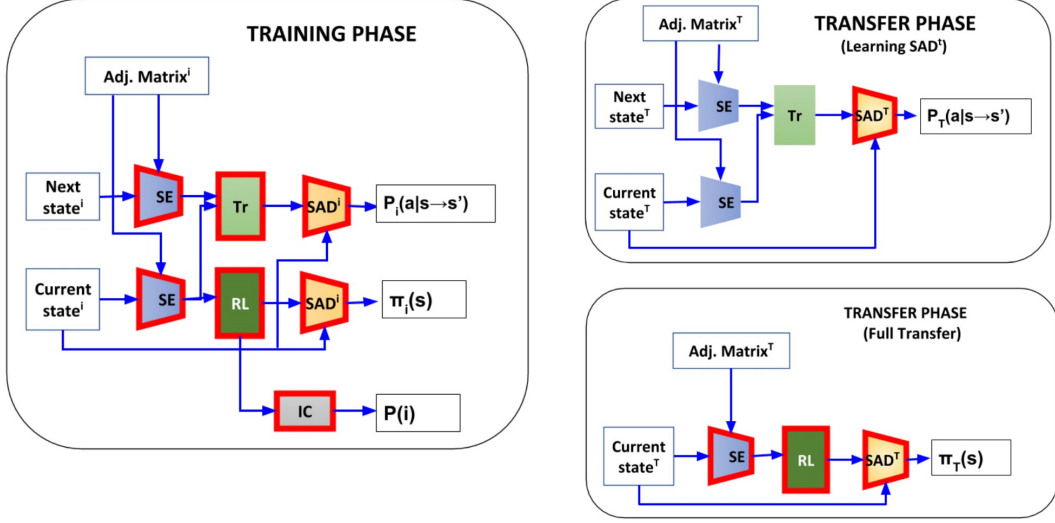

Figure 1: Model architecture for TORPIDO. The architecture is divided into three phases: training phase, transfer phase (learning $SAD^T$) and transfer phase (full transfer). The figure shows training of $i^{th}$ problem – this is replicated $N$ times with shared SE, Tr, RL and IC modules. The same color boxes have same weights during each step. Across different steps the same color boxes signify that they have been initialized from previous step. The red color outline signifies that those weights are being trained in that step.

and RL module share both the state and state-action embedding spaces. This novel module allows us to quickly learn an action decoder, thus allowing a near-zero shot transfer to take place.

**Instance classifier:** TORPIDO's aim is to learn state-action embeddings *independent* of the instance (since they are shared between all instances). To explicitly enforce this condition, we use an idea from domain adaptation [Ganin et al., 2017]. Essentially, as an auxiliary task, we try to learn a classifier to predict the problem instance, given a state-action embedding. This is done in an adversarial manner, so that the model learns to produce such state-action embeddings, that even the best possible classifier is unable to predict the instance from which they were generated. In the ideal case, at equilibrium, the model learns to produce state-action embeddings which are instance invariant, as even the best instance classifier would predict an equal probability over all source instances.

### 4.1 TORPIDO's Training & Transfer Phases

**Learning Phase:** During the learning phase, we learn a policy over each of the $N$ problem instances in a multi-task manner by sharing the state encoder and RL module, and by using separate decoders for each instance. We also learn the transition function to predict the distribution of actions given consecutive states. The instance invariance is implemented using a gradient reversal layer, as in [Ganin et al., 2017]. I.e., the gradients for the instance classification loss are back-propagated in the standard manner through the instance classifier layer, but with their sign reversed in all the layers preceding the state-action embedding (hence enforcing the adversarial objective function of the game described). The loss function for training is a weighted sum of the policy gradient loss of A3C, a cross-entropy loss for prediction of the actions given consecutive states (from the transition module), and the instance misclassification loss, i.e., the cross entropy loss from the instance module with sign reversed. The instance classification module is trained to minimize the cross-entropy loss between the predicted instance distribution and ground-truth instance. Mathematically, $E(\theta_E, \theta_{D1}, ..., \theta_{DN}, \theta_{IC}, \theta_T) =$

$$\sum_{i=1}^{N} L_p(\theta_E, \theta_{Di}) - \lambda \sum_{i=1}^{N} L_c(\theta_E, \theta_{IC}) + \lambda_{tr} \sum_{i=1}^{N} L_{tr}(\theta_E, \theta_{Di}, \theta_{tr}) \tag{1}$$

where $N$ is the number of training instances, $L_p$ is the loss function of the policy network of the A3C agent, $L_c$ is the cross-entropy loss for the instance classifier and $L_{tr}$ is cross-entropy loss for transition module. Here $\theta_E$ represent the combined parameters of the encoder and RL module, $\theta_{Di}, i = 1...N$ represents the parameters of the decoder module of the $i^{th}$ agent, $\theta_{IC}$ represents the parameters of the instance classifier module and $\theta_{tr}$ represents parameters of tran-

Table 1: Comparison of TORPIDO against other baselines, stopped at 4 different iteration numbers.

| Train iter | 0 | | | 0.1M | | | 0.5M | | | ∞ | | |
|---|---|---|---|---|---|---|---|---|---|---|---|---|
| Algo | A3C | A2T | TP | A3C | A2T | TP | A3C | A2T | TP | A3C | A2T | TP |
| Sys#1 | 0.00 | 0.08 | 0.23 | 0.01 | 0.09 | 0.26 | 0.11 | 0.19 | 0.39 | 0.31 | 0.38 | 1.0 |
| Sys#5 | 0.00 | 0.02 | 0.26 | 0.03 | 0.11 | 0.30 | 0.08 | 0.17 | 0.64 | 0.30 | 0.40 | 1.0 |
| Sys#10 | 0.02 | 0.06 | 0.26 | 0.04 | 0.09 | 0.33 | 0.08 | 0.13 | 0.49 | 0.32 | 0.33 | 1.0 |
| Game#1 | 0.00 | 0.19 | 0.34 | 0.04 | 0.22 | 0.49 | 0.43 | 0.60 | 0.98 | 0.54 | 0.61 | 1.0 |
| Game#5 | 0.00 | 0.03 | 0.41 | 0.11 | 0.11 | 0.55 | 0.24 | 0.17 | 0.77 | 0.40 | 0.44 | 1.0 |
| Game#10 | 0.07 | 0.05 | 0.34 | 0.03 | 0.08 | 0.49 | 0.08 | 0.14 | 0.88 | 0.20 | 0.20 | 1.0 |
| Navi#1 | 0.00 | 0.04 | 0.72 | 0.01 | 0.04 | 0.72 | 0.10 | 0.19 | 0.9 | 0.22 | 0.25 | 1.0 |
| Navi#2 | 0.00 | 0.01 | 0.68 | 0.05 | 0.06 | 0.73 | 0.26 | 0.45 | 1.0 | 0.55 | 0.56 | 1.0 |
| Navi#3 | 0.00 | 0.01 | 0.50 | 0.03 | 0.03 | 0.60 | 0.21 | 0.42 | 0.71 | 0.40 | 0.40 | 1.0 |

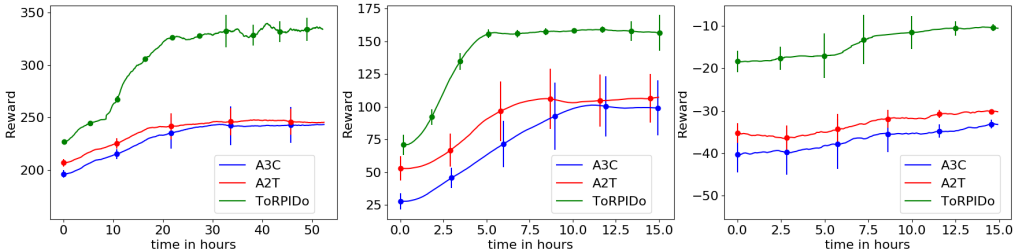

Figure 2: Learning curves on 1st problem of the three domains: (a) SysAdmin, (b) Game of Life, (c) Navigation. In all cases TORPIDO outperforms other baselines by wide margins.

sition module. We are seeking parameters $\theta_E^*$, $\theta_{Di}^*$, and $\theta_{IC}^*$ that deliver a saddle point of the functional such that $(\theta_E^*, \theta_{D1}^*, ..., \theta_{DN}^*) = argmin_{\{\theta_E, \theta_{D1}, ..., \theta_{DN}\}} E(\theta_E, \theta_{D1}, ..., \theta_{DN}, \theta_{IC})$ and $\theta_{IC}^* = argmax_{\{\theta_{IC}\}} E(\theta_E, \theta_{D1}, ..., \theta_{DN}, \theta_{IC})$.

**Transfer Phase:** During this phase, the state encoder requires simply inputting the adjacency matrix of the target instance and it directly outputs state embeddings for this problem using SE. Since our RL agent operates in the embedding space, it is exactly the same for the target instance, and, hence, is directly transferred. However, an action decoder needs to be relearnt. For this, we make use of the fact that the transition function also operates only in the embedding space, so can also be directly transferred. For the target instance, we try to predict the distribution over actions given consecutive states in the new instance. The weights for the state encoder and transition module remain fixed, while the weights for the decoder for the new instance are learned. This decoder can then be directly used to transform the state-action embeddings predicted by the RL module into a distribution over actions, or a policy for the new instance. Hence, we are able to achieve a near zero-shot transfer, i.e., without doing any RL in the new environment and by simply retraining action encoder via transition transfer. After the weights of each module have been initialized as above, TORPIDO follows the same training procedure as in A3C. This generates the learning curve, post the zero-shot transfer.

In summary, this architecture leverages the extra information in RDDL domains in two ways: (i) it uses the input structure to represent the state as a graph, and uses a GCN to learn a state embedding (ii) it uses the transition function (available in the RDDL file) to learn the decoder for the new domain.

# 5  Experiments

We wish to answer three experimental questions. (1) Does TORPIDO help in transfer to new problem instances? (2) What is the comparison between TORPIDO and other state of the art transfer learning frameworks? (3) What is the importance of each component in TORPIDO?

**Domains:** We make all comparisons on three different RDDL domains used in IPPC, International Planning Competition 2014 [Grzes et al., 2014] – SysAdmin, Game of Life and Navigation. We have already described the SysAdmin domain [Guestrin et al., 2001] in background section. The Game of Life represents a grid-world cellular automata, where each cell is dead or alive. Each alive cell continues to live in next step as long as there is no over- or under-population (measured by number

Table 2: Incremental value of each component of TORPIDO at the start of the training (zero-shot).

|  | A3C+GCN | A3C+GCN + SAD | A3C+GCN + SAD + IC |
|---|---|---|---|
| Sys#1 | 0.01 | 0.25 | 0.23 |
| Sys#5 | 0.01 | 0.23 | 0.26 |
| Sys#10 | 0.01 | 0.22 | 0.26 |
| Game#1 | 0.03 | 0.31 | 0.34 |
| Game#5 | 0.02 | 0.26 | 0.41 |
| Game#10 | 0.01 | 0.24 | 0.34 |
| Navi#1 | 0.05 | 0.70 | 0.72 |
| Navi#2 | 0.03 | 0.59 | 0.68 |
| Navi#3 | 0.01 | 0.27 | 0.50 |

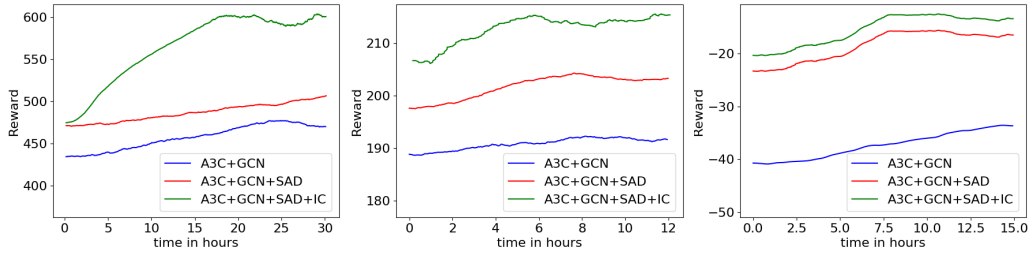

Figure 3: Incremental value of various components of TORPIDO. Results on 2nd problem of the three domains: (a) SysAdmin, (b) Game of Life, (c) Navigation. In all cases state-action decoder is critical for good performance. The instance classifier has marginal benefits.

of adjacent live cells). Additionally, the agent can make any one cell alive in each step. Finally, the Navigation domain requires a robot to move in a grid from one location to the other using four actions – up, down, left and right. There is a river in the middle and the robot can drown with a non-zero probability in each location. If it drowns, it restarts from the start state in the next episode and current episode is over.

All three of these domains have some spatial structure, but that is implicit in the symbolic description (exposed via non-fluents in RDDL). This makes them ideal choices for a first study of its kind. For each domain we perform experiments on three different instances. A higher numbered problem roughly corresponds to a problem with much larger state space. For example, SysAdmin1 has 10 computers and SysAdmin10 has 50 computers (effective state space sizes $2^{10}$ and $2^{50}$ respectively).

**Experimental Settings and Comparison Algorithms:** In the spirit of domain-independent planning, all hyperparameters are kept constant for all problems in all domains. Our parameters are as follows. A3C's value network as well as policy network use two GCN layers (3, 7 feature maps) and two fully connected layers. The action decoder implements two fully connected layers. All layers use the exponential linear unit (ELU) activations [Clevert et al., 2015]. All networks are trained using RMSProp with a learning rate of $5e^{-5}$. For TORPIDO, we set $N = 4$, i.e., the training phase uses four source problems. Random problems are generated for training using the generators available for each domain. All weights of GCN of policy network, and the RL module of policy network are shared.

We implement two baseline algorithms for comparison. First, we implement a base non-transfer algorithm, A3C. This is chosen since TORPIDO uses A3C as its RL agent. Thus, this comparison will directly show the value of transfer. We also implement a state of the art RL transfer solution called A2T – Attend, Adapt and Transfer [Rajendran et al., 2017], which retrains while using attention over the learned source policies for selective transfer. It also uses the same four problems as source as in TORPIDO. This comparison will expose the specific value of our transfer mechanism, which uses the RDDL representation, compared to a representation-agnostic transfer mechanism.

**Evaluation Metrics:** First, we output learning curves. For that we stop training after a set number of training iterations (say $i$) and estimate the return from the current policy $V_\pi(i)$ by simulating it till the horizon specified in the RDDL file. The reported values are an average of 100 such simulations. Moreover, we also report the metric $\alpha(i)$, where $\alpha(i) = (V_\pi(i) - V_{inf})/(V_{sup} - V_{inf})$. Here $V_{inf}$ and $V_{sup}$ respectively represent the lowest and the highest values obtained on this instance (by any

algorithm at any time in training). Since $\alpha$ is a ratio, it acts as an indicator of the training 'stage' of the model, and hence helps to understand the transfer process as it progresses in time, irrespective of the starting (random) reward and the final reward for the model. Moreover, $\alpha(0)$ acts as a measure of (near) zero-shot transfer.

## 5.1  TORPIDO's Transfer Ability

We first measure the ability of the model to transfer knowledge across problem instances. We compare against all baselines. Figure 2 compares the learning curves of TORPIDO with the two baselines on one problem each of the three domains (error bars are 95% confidence intervals over ten runs). The results on the other problems are quite similar. The x-axis is RL training time on the target instance, which for TORPIDO also includes the time for training of action decoder. First, we find that A3C is not very competitive with even A2T in its learning. This suggests that transfer is quite valuable for these problems. We also find that A2T itself has substantially worse performance compared to TORPIDO. We attribute this to the various components in TORPIDO that exploit the domain knowledge expressed in RDDL representation.

In Table 1 we show the comparisons between these algorithms at four different training points. We report the $\alpha$ values, as described above. We find that TORPIDO is vastly ahead of all algorithms at all times, underscoring the immense value our architecture offers.

## 5.2  Ablation Study

In order to understand the incremental contribution of each of our components we compare three different versions of TORPIDO. The first version is A3C+GCN. This version only performs state encoding but does not perform any action decoding. Our next version is A3C+GCN+SAD, which incorporates the action decoding (and also transition transfer module to aid action decoding). Finally, our full system adds an IC to previous name – it includes the instance classification component.

Figure 3 shows the learning curves for the three problems. We observe that use of GCN helps the algorithm converge to a high final score. Comparing this to vanilla A3C and A2T in Figure 2, we learn that the use of GCN is critical in exposing the structure of the domain to the RL agent, helping it in learning a final good policy. However, the zero-shot nature of the transfer is very weak, because the action names may be different in the source and target. Use of action-decoder and transition transfer speeds up the near zero-shot transfer immensely. This can be observed from Table 2, which compares these algorithms before starting the RL training. We see a huge jump in $\alpha$ for the model with action decoder compared to the one without it. Finally, Table 2 suggests that the improvement of instance classification in the beginning is significant. However, very soon the incremental benefit is reduced; final TORPIDO performs only marginally better than the A3C+GCN+SAD version.

## 6  Conclusions

We present the first domain-independent transfer algorithm for transferring deep RL policies from source probabilistic planning problems (expressed in RDDL language) to a target problem from the same domain.[2] Our algorithm TORPIDO combines a base RL agent (A3C) with several novel components that use the RDDL model: state encoder, action decoder, transition transfer module and instance classifier. Only action decoder needs to be re-learnt for a new problem; all the other components can be directly transferred. This allows TORPIDO to perform an effective transfer even before the RL starts, by quickly retraining the action decoder using the given RDDL model (near zero-shot learning). Experiments show that TORPIDO is vastly superior in its learning curves compared to retraining from scratch as well as a state-of-the-art RL transfer method. In the future, we wish to extend this to transfer across problem sizes, and later transfer across domains.

## Acknowledgements

We thank Ankit Anand and the anonymous reviewers for their insightful comments on an earlier draft of the paper. We also thank Alan Fern, Scott Sanner, Akshay Gupta and Arindam Bhattacharya

for initial discussions on the research. This work is supported by research grants from Google, a Bloomberg award, an IBM SUR award, a 1MG award, and a Visvesvaraya faculty award by Govt. of India. We thank Microsoft Azure sponsorships, and the IIT Delhi HPC facility for computational resources.

## Footnotes

[1]Available at `https://github.com/dair-iitd/torpido`

[2]Available at `https://github.com/dair-iitd/torpido`

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
