[Reviews · NeurIPS 2018]

Reviewer 1



The authors propose a system that is composed of a state encoder (using a graph convolutional network and assuming object and relational information from the environment). They learn a policy in the representation space that outputs a state-action pair embedding, an adversarial instance classifier to try to learn domain independent embeddings, and also have an action decoder for each task and a transition transfer module to ease transfer. Does your baseline A3C include using a GCN? They do provide ablation studies in Table 2, which show that the instance classifier (IC) helps with zero-shot transfer. I don't understand Figure 3? It looks like none of the additional pieces on top of GCN improve performance on the first task -- which is not surprising, so not sure why this figure is here. Transfer experiments: There is no detail on the transfer experiments. How many samples do you use to train the action decoder and transition transfer function between tasks? Is the point being made that these samples "don't count" towards transfer because they don't exploit the reward function, only the dynamics of the new task? The proposed method is promising but I want more information about the transfer experiments.

Reviewer 2



The paper proposes a method termed TransPlan, to use Graph Convolutional Networks to learn the relations defined by an RDDL description to learn neural network policies that can "transfer" to different MDP planning domain instances. The architecture combines several components including a state encoder, an action decoder, a transition transfer module and a problem instance classifier. Only the action decoder requires retraining for transfer and the paper shows how a different component in the architecture (transition transfer module) can be used to quickly retrain and get substantial gains in transfer to a new domain without any "real" interactions (zero-shot). Authors evaluate their performance on benchmark domains from IPPC 2014 and show substantial improvements over standard algorithms which do not leverage the structure offered by an RDDL description of the problem. The authors also a do a few ablations studies to find the relative importance of different components in their system. # Strengths The paper proposes a method termed TransPlan, to use Graph Convolutional Networks to learn the relations defined by an RDDL description to learn neural network policies that can "transfer" to different MDP planning domain instances. The architecture combines several components including a state encoder, an action decoder, a transition transfer module and a problem instance classifier. Only the action decoder requires retraining for transfer and the paper shows how a different component in the architecture (transition transfer module) can be used to quickly retrain and get substantial gains in transfer to a new domain without any "real" interactions (zero-shot). Authors evaluate their performance on benchmark domains from IPPC 2014 and show substantial improvements over standard algorithms which do not leverage the structure offered by an RDDL description of the problem. The authors also a do a few ablations studies to find the relative importance of different components in their system. # Weakness: Which brings us to various unclear parts in the paper. First of all, there are key claims that are hard to justify. For instance: "a key strength of neural models is their effectiveness at efficient transfer". I am sure you'll find a lot of disagreement here especially when you are not working in the vision community where ImageNet models transfer fairly well. This is not the case with models trained with RL (DQN or policy gradients) because the gradients are a lot noisier and the representations learning is more difficult. Another example being, "A3C algorithm […], because it is simple, robust and stable" is hard to swallow given the term "asynchronous" in the algorithm. The paper's treatment of "transfer" is quite unclear too. Transfer learning has a long history and would refer to multiple surveys on transfer in RL [1,2] to better place their objective. Moreover, we can expect that the NIPS audience wouldn't know as much about symbolic AI and RDDL description, so use of terms like fluents without defining them first, leaves things unclear to the reader. Similarly, even thought the components of the architecture are clearly explained individually, their exact combination and how exactly the losses are setup is quite unclear. I hope the authors are atleast planning on releasing their code for easier replication. There are quite a few components in the proposed method. Whether they are warranted can only be checked by experimental verification. The paper is quite unclear about the exact nature of these problem domains - what's the observation space like, what are the possible number of problems that are generated in a domain, etc. (One can look it up on IPPC but making the paper clear would be better). Moreover, since these are planning problems it's hard to say if DeepRL algorithms like A3C are right baselines to benchmark against. The paper _is_ using the model after all (transition). Wouldn't it be useful to at least show the standard methods used in IPPC and their efficiency. Do we even gain on anything at all by showing transfer abilities if the total time taken by standard planning algorithms for each problem domain is less than learning via DeepRL? Moreover it's unclear what were the parameters of A3C algorithms itself - number of workers/batch size etc. It doesn't look like Figure 2 and Figure 3 show averaged runs over multiple seeds (maybe fill color to show standard error?) nor is there any standard deviation for Table 2 results. So although the inclusion of "SAD" seems pretty big as an improvement, I can't make a strong claim given how much variance there can be with DeepRL experiments. # Minor things: - Line 85: Missing citation? - Line 329: Not sure if Table 2 suggests that. Possibly meant to be Figure 3? [1]: http://www.jmlr.org/papers/volume10/taylor09a/taylor09a.pdf [2]: https://hal.inria.fr/hal-00772626/document

Reviewer 3



(i) Summary: The paper proposes TransPlan, a novel neural architecture dedicated to “near” zero-shot transfer learning between equi-sized discrete MDP problems from the same RDDL domain (i.e., problems with same number of state and action variables, but with different non-fluents / topologies). It combines several deep neural models to learn state and state-action embeddings which shall be used to achieve sample-efficient transfer learning across different instances. It uses GCNs (Graph Convolutional Networks), a generalization of ConvNets for arbitrary graph embeddings, to learn state embeddings, A3C (Asynchronous Advantage Actor Critic) as its RL module that learns policies mapping state embeddings to an abstract representation of probability distributions over actions, and action decoders that attempt to transform these abstract state-action embeddings into the MDP’s original action space for the target problem. Experiments on IPPC domains (e.g., SysAdmin, Game of Life, and Navigation) compare the proposed approach with A3C (without transfer learning) and A2T (Attend, Adapt and Transfer). Results indicate that TransPlan can be effective for transfer learning in these domains. (ii) Comments on technical contributions: - Contributions seem a bit overstated: The authors argue that the proposed transfer learning approach is the “first domain-independent transfer algorithm for MDP planning domains expressed in RDDL”. Regardless of whether TransPlan is the first transfer algorithm for the RDDL language, the proposed approach does not seem “domain-independent” in a very general sense, given that its state encoder relies on RDDL domains exhibiting a graph structure (i.e., “state variables as nodes, edges between nodes if the respective objects are connected via the non-fluents in the domain”) - which explains the particular choice of IPPC domains for experiments. Moreover, the transfer learning is only focused on target instances with same size as the instances used in the learning phase, which seems like an important limitation. Finally, it is not clear until the experiments section that TransPlan is only applicable to discrete domains, another limitation to a subset of RDDL. - The paper contributions are not put in context with recent related work: The main works related to the idea of Deep Reactive Policies in model-known planning (e.g., “Action schema networks: Generalised policies with deep learning”, and “Training deep reactive policies for probabilistic planning problems”) are barely described and not discussed at all. In particular, it is not even mentioned that Action-Schema Nets also address the transfer learning between instances of the same model independently of the size of the target instances. Is the RDDL vs. PPDDL issue that important that these methods are incomparable? - The background necessary to appreciate the proposed method could be presented better. Section 2.1. Reinforcement Learning The presentation of the Actor-Critic architecture (which is the core aspect of A3C) is really confusing. First off, it is said that A3C “constructs approximations for both the policy (using the ‘critic’ network) and the value function (using the ‘actor’ network)”. This is wrong or at least very unclear. The actor network is the approximation for the policy and the critic network it is used as the baseline in the advantage function (i.e., the opposite of what the authors seem to say). Moreover, the formulation of the policy gradient seems totally wrong as the policy \pi(a|s) is the only \theta-parameterized function in the gradient (it should be denoted instead as \pi_{\theta}(a|s)), and also the value functions Q_{\pi}(s, a; \theta) and V(s; \theta) do not share the same set of parameters - indeed, Q-values are approximated via Monte-carlo sampling and V(s; \theta) is the baseline network learned via regression. Additionally, the actor network does not maximize the H-step lookahead reward by minimizing the expectation of mean squared loss, the minimization of the mean squared loss is part of the regression problem solved to improve the approximation of the critic network (i.e., the baseline function); it is the policy gradient that attempts to maximize the expected future return. Section 2.2. Probabilistic Planning The presentation of the RDDL language is not very clear and could mislead the reader. Aside from technical imprecision, the way it is presented can give the reader the impression that the language is supposed to be solely oriented to “lifted” MDPs, whereas I understand that it is a relational language aimed at providing a convenient way to compactly represent ground factored MDPs problems. This intent/usage of RDDL should be clarified. Section 2.3 Graph Convolutional Networks - The formulas are not consistent with the inputs presented: The explanation of the inputs and outputs of the GCN layer in lines 149-151 are confusing since parameter N is not explained at all and D^{(l+1)} is simply not used in the formula for F^{(l+1)}. - The intuition for the GCN layer and propagation rule/formula is too generic: The formula for the GCN layer’s activation function has not meaning at all for anyone not deeply familiar with ICLR2017 paper on Semi-supervised classification with graph convolutional networks. Also, the intuition that “this propagation rule implies that the feature at a particular node of the (l + 1)th layer is the weighted sum of the features of the node and all its neighbours at the lth layer” is completely generic and not does not elucidate at all the formula in line 152. Finally, the statement that “at each layer, the GCN expands its receptive field at each node by 1” is not explained and difficult to understand for someone not expert in GCNs. - Problem formulation is not properly formalized: The authors say that “we make the assumption that the state and action spaces of all problems is the same, even though their initial state, non-fluents and rewards could be different”. Putting together this statement with the following statement in lines 95-96 “Our work attempts a near-zero shot learning by learning a good policy with limited learning, and without any RL.” triggers some questioning not explicitly addressed in the paper about how could the rewards be different in the target instances and the transfer learning still be effective without any RL. The authors should really clarify what they mean by the “rewards could be different”. (iii) Comments on empirical evaluation: The empirical evaluation is centered around answering the following questions: (1) Does TRANSPLAN help in transfer to new problem instances? (2) What is the comparison between TRANSPLAN and other state of the art transfer learning frameworks? (3) What is the importance of each component in TRANSPLAN? Questions (1) and (2) are partially answered by the experiments. Indeed, TransPlan does seem to transfer the learning to new problem instances (given the limitations pointed earlier) and it improves on the transfer learning A2T approach. But these conclusions are based solely on “learning curves” plotted in terms of “number of iterations”. So, it is important to remember that the first and foremost motivation of transfer learning is to amortize the computational cost of training the neural model over all target instances of the domain. If this is not successfully accomplished, there is not point in incurring the cost of the offline learning phase, and then it is best to plan from scratch for each instance of the domain. Therefore, it is in my understanding that a better experimental design should focus on measuring and comparing learning/training times and transfer times, instead of relying on the number of iterations to showcase the learning evolution of TransPlan. Particularly, this is important to fairly highlight the value of transfer learning when comparing with baseline A3C, and the particular advantages of TransPlan when comparing with the model-free A2T transfer learning approach. Additionally, the authors should better explain why in Table 1 the columns for iter 0 are not 0.0 and the last columns for iter \infty are not 1.0. Judging by the formula of \alpha(i) in line 300, this should be the case. (iv) Detailed comments: Typos: - Line 27: “[...] and RDDL planning solve each problem instance [...]” should be “[...] and RDDL planning that solve each problem instance [...]” - Line 85: it lacks the proper reference for “transferring the value function or policies from the source to the target task []” - Line 109: “RDDL Reprentation” should be “RDDL Representation” Other suggestions for better clarity: - Lines 107-108: “[...] an MDP exponential states [...]” should be “[...] an MDP exponential state space [...]” - Line 123: “[...] this domain is inherently non-deterministic [...]” would be better phrased “[...] this domain is inherently dynamic [...]”? - Figure 1: The model architecture is not really clear. Perhaps it would be better if the neural components used in the training/learning and transfer phases were separated with its inputs and outputs clearly shown in the figure. Also, an algorithm detailing the training procedure steps would greatly improve the presentation of the TransPlan approach. (v) Overall evaluation: The paper is relevant for the NIPS audience. It brings together interesting and novel ideas in deep reactive networks and training methods (i.e., domain-adversarial training). However, the overall readability of the paper is compromised by the unclear background section, the ambiguous presentation of the model architecture, and technical issues with some formulae. Additionally, the experimental setup seems biased to showcase what is stated by the authors in lines 314 - 315 “TRANSPLAN is vastly ahead of all algorithms at all times, underscoring the immense value our architecture offers to the problem.” In my understanding, it is a major experimental drawback to not compare the proposed approach with the baseline approaches w.r.t. the training/transfer times (instead, all comparisons are based on number of iterations), which is common in RL and transfer learning testbeds. So, my recommendation is to reject the paper mainly on the premises that the overall presentation must be improved for clarity/correctness; ideally the experiments should also take into consideration the computational costs in terms of training and transfer times.